# Low CDK Activity and Enhanced Degradation by APC/C^CDH1^ Abolishes CtIP Activity and Alt-EJ in Quiescent Cells

**DOI:** 10.3390/cells12111530

**Published:** 2023-06-01

**Authors:** Fanghua Li, Emil Mladenov, Yanjie Sun, Aashish Soni, Martin Stuschke, Beate Timmermann, George Iliakis

**Affiliations:** 1Institute of Medical Radiation Biology, University Hospital Essen, University of Duisburg-Essen, 45147 Essen, Germany; fanghua.li@uk-essen.de (F.L.); emil.mladenov@uk-essen.de (E.M.); yanjie.sun@uk-essen.de (Y.S.); aashish.soni@uk-essen.de (A.S.); 2Department of Particle Therapy, University Hospital Essen, West German Proton Therapy Centre Essen (WPE), West German Cancer Center (WTZ), German Cancer Consortium (DKTK), 45147 Essen, Germany; beate.timmermann@uk-essen.de; 3Division of Experimental Radiation Biology, Department of Radiation Therapy, University Hospital Essen, University of Duisburg-Essen, 45147 Essen, Germany; martin.stuschke@uk-essen.de; 4German Cancer Consortium (DKTK), Partner Site University Hospital Essen, German Cancer Research Center (DKFZ), 45147 Essen, Germany

**Keywords:** ionizing radiation, DNA repair, repair of DNA double-strand breaks, alt-EJ, DNA end-resection, CtIP, APC/C, CDKs, RPA, pulsed-field gel electrophoresis

## Abstract

Alt-EJ is an error-prone DNA double-strand break (DSBs) repair pathway coming to the fore when first-line repair pathways, c-NHEJ and HR, are defective or fail. It is thought to benefit from DNA end-resection—a process whereby 3′ single-stranded DNA-tails are generated—initiated by the CtIP/MRE11-RAD50-NBS1 (MRN) complex and extended by EXO1 or the BLM/DNA2 complex. The connection between alt-EJ and resection remains incompletely characterized. Alt-EJ depends on the cell cycle phase, is at maximum in G_2_-phase, substantially reduced in G_1_-phase and almost undetectable in quiescent, G_0_-phase cells. The mechanism underpinning this regulation remains uncharacterized. Here, we compare alt-EJ in G_1_- and G_0_-phase cells exposed to ionizing radiation (IR) and identify CtIP-dependent resection as the key regulator. Low levels of CtIP in G_1_-phase cells allow modest resection and alt-EJ, as compared to G_2_-phase cells. Strikingly, CtIP is undetectable in G_0_-phase cells owing to APC/C-mediated degradation. The suppression of CtIP degradation with bortezomib or CDH1-depletion rescues CtIP and alt-EJ in G_0_-phase cells. CtIP activation in G_0_-phase cells also requires CDK-dependent phosphorylation by any available CDK but is restricted to CDK4/6 at the early stages of the normal cell cycle. We suggest that suppression of mutagenic alt-EJ in G_0_-phase is a mechanism by which cells of higher eukaryotes maintain genomic stability in a large fraction of non-cycling cells in their organisms.

## 1. Introduction

Exposure of cells to ionizing radiation (IR) induces DNA double-strand breaks –(DSBs) that challenge genomic stability. In higher eukaryotes, DSBs are mainly processed by classical non-homologous end-joining (c-NHEJ) and homologous recombination (HR), while alternative end-joining (alt-EJ) and single-strand annealing (SSA) contribute variably depending on several parameters, including the cell cycle phase and growth state [1,2]. While HR is conceptually designed to faithfully restore the genome [2], c-NHEJ effectively rejoins DNA ends without template requirements and is, therefore, less faithful than HR. However, its simplicity and speed make it the most prominent route of DSB repair in higher eukaryotes, especially after high doses of IR exposure [3,4,5,6]. The failures of c-NHEJ or HR may be compensated by alt-EJ [6,7,8], which is slow, more error-prone than c-NHEJ, and, therefore, a frequent source of chromosomal translocations. This defines alt-EJ as a mutagenic route of DSB processing that promotes genomic instability induction [9].

How c-NHEJ and HR suppress error-prone alt-EJ to maintain genomic stability is a subject of intensive research. However, emphasis is presently placed on the regulation of DNA end-resection (henceforth here simply, resection). Resection processes the end of a DSB to expose a 3′ single-stranded DNA that can then invade a homologous DNA molecule or anneal with another resected end using homologies. Resection inhibits c-NHEJ and triggers homology-directed DSB repair pathways, including HR and SSA, that play key roles in the DSB repair pathway choice. Lower degrees of homology are utilized by alt-EJ [10]. Resection requires CtIP [11] to stimulate the endonuclease activity of MRE11 and proceeds bi-directionally [12,13]. The activity of the MRE11-RAD50-NBS1 (MRN) complex processes the DNA end in the 3ʹ–5ʹ direction, while Exonuclease 1 (EXO1) or Bloom helicase (BLM)/DNA Replication Helicase/Nuclease 2 (DNA2) catalyze long-range 5ʹ–3ʹ resection [14]. Resection is regulated in a cell-cycle-dependent manner and is maximally active in G_2_-phase cells, with significantly reduced levels in G_1_ [15,16].

Alternative end-joining (alt-EJ) still remains incompletely characterized, and it is widely accepted that it functions by default when KU- and DNA-dependent serine/threonine protein kinase catalytic subunits (DNA-PKcs), Ligase 4 (LIG4) or other c-NHEJ factors are defective [17,18,19,20]. Alt-EJ may make greater use of microhomology (MH) compared to c-NHEJ [21,22,23]. However, it remains questionable whether MH-dependency is a universal requirement for all alt-EJ events at all conditions. Indeed, by studying the utilization of homologies in c-NHEJ-proficient and -deficient cells, Mansour demonstrated that alt-EJ was MH-independent [24]. In conclusion, alt-EJ was defined as any c-NHEJ-independent end-joining process, and dependence on MH has been well characterized, but it may not always be required. Based on these findings, two kinds of systems of detection and characterization of alt-EJ were established: residual repair of IR-induced DSBs after c-NHEJ factor deactivation or the presence of MH at junctions generated in model DSBs—typically enzymatically induced.

Adopting the concept of MH detection, appropriately designed reporter assays monitoring GFP-expression after processing of *I-Sce I*-induced DSBs by a specific DSB repair pathway are frequently used to characterize alt-EJ [10,25,26]. Such studies have reported that MRE11 is required for the resection of as few as 20 bp during alt-EJ [10], and similar conclusions have been drawn for CtIP and NBS1 [10,19,27,28,29]. However, in these reporter assays, microhomology is built-in, which may result in an overestimation of the contributions of resection to alt-EJ.

Measuring DSBs directly, using pulsed-field gel electrophoresis (PFGE), allows the study of DSB induction and repair kinetics, and in these experiments residual repair activity after c-NHEJ inactivation is taken to reflect the contribution of alt-EJ. Using PFGE, we have recently shown that CtIP depletion suppresses alt-EJ in both exponentially growing and in G_2_-phase-enriched cells [30]. One key characteristic of alt-EJ uncovered using PFGE is its crucial dependence on the state of growth. Thus, alt-EJ activity is low during G_1_-phase and steadily increases as cells move to G_2_-phase. Notably, alt-EJ is almost completely abrogated in some cell lines in G_0_-phase [31,32,33]. Epidermal growth factor receptor (EGFR) inhibition in proliferating cells suppresses alt-EJ only marginally, and the addition of EGF in G_0_-phase cells increases alt-EJ marginally, which suggests that the inhibition of alt-EJ in G_0_ cells is not a direct consequence of suppressed growth factor signaling [34]. While the mechanism underlying the suppression of alt-EJ in G_0_-phase cells still remains to be elucidated, a tenable hypothesis is that suppression of alt-EJ derives from a further reduction in resection in G_0_, which immediately raises the question as to the underpinning mechanism.

The cell-cycle-dependent regulation of resection is achieved through CDKs [35]. CtIP phosphorylation by CDKs on Thr847 and Ser327 critically regulates resection in vertebrates [36,37,38]. Interestingly, CDK2-dependent phosphorylation of Ser276 and Thr315 also promotes CtIP binding to PIN1 to dampen resection [39]. CDKs also promote resection by phosphorylating EXO1 [40], NBS1 [41,42,43] and DNA2 [44]. Finally, CDKs regulate resection by suppressing resection blocks raised by 53BP1 and DNA helicase B (HELB) [45,46,47].

The oscillating activity of CDKs throughout the cell cycle is regulated by the periodic degradation, via the ubiquitin–proteasome system, of cyclins and CDK inhibitors (CKIs) [48]. Central in this process is a pair of RING-type E3 ubiquitin ligases: SCF (for SKP1/Cullin/F-box protein) and the anaphase-promoting complex/cyclosome (APC/C) [49]. SCF remains active from late G_1_- to late G_2_-phase and selectively degrades proteins through priming for degradation. S-phase kinase-associated protein 2 (SKP2) is the main substrate recognition factor of SCF [49]. CDC20 homolog-1 (CDH1) is the main substrate recognition factor of the APC/C complex, which is activated in late M- to early/mid-G_1_-phase and retains activity until the inception of DNA replication [49,50].

Notably, in addition to CDKs, APC^CDH1^ also regulates resection by regulating cyclins (and, thus, indirectly CDK activity) and CtIP abundance. IR activates APC^CDH1^ and mediates CtIP degradation to limit resection in G_2_-phase cells [51]. In addition, APC^CDH1^ competes with SCF, and indeed, SKP2 depletion causes the hyper-activation of APC^CDH1^, and thus, the full degradation of CtIP that compromises alt-EJ in G_2_-phase-enriched cells [30]. These results suggest that the regulation of alt-EJ is achieved through CDK-dependent activation of key components of the resection apparatus and the SCF-APC/C-dependent regulation of their levels by protein degradation. The regulation of alt-EJ by CtIP levels and its CDK-dependent activation remains to be elucidated in G_1_- and, particularly, in G_0_-phase cells.

In the present study, we continue with the elucidation of the mechanistic regulation of alt-EJ throughout the cell cycle by focusing on cells in G_1_- and G_0_-phase. While notable exceptions have been reported [52,53], HR is generally thought to be suppressed in this phase of the cell cycle. The same holds true for SSA, which is known to require longer resection than HR. These characteristics of repair pathway activity in G_1_/G_0_-phase cells make it possible to, rather specifically, investigate the role of resection in alt-EJ [10,54,55,56] by simply inhibiting c-NHEJ—the only DSB repair pathway that retains detectable activity in these cells.

We report that CtIP is required for alt-EJ activity in G_1_-phase and that its degradation or reduced phosphorylation abrogates resection and alt-EJ. Interestingly, G_0_-phase cells are more radioresistant to killing than G_1_-phase cells, which suggests that the suppression of alt-EJ can be beneficial for the cell. Since almost all terminally differentiated, post-mitotic cells in the human body are non-cycling, and because unstimulated stem cells frequently rest in G_0_-phase, it is possible that the suppression of error-prone alt-EJ is a natural mechanism safeguarding genomic stability and protecting against tumorigenesis.

## 2. Materials and Methods

### 2.1. Cell Culture and Irradiation

Normal, immortalized human fibroblasts, 82-6 hTert (provided by Drs. M. Lobrich and P. Jeggo) [16]; and human lung carcinoma cells, A549 (ATCC, CCL-185) and its derivatives, *CDH1^−/−^* [30] and *POLQ^−/−^*; human osteosarcoma cells, U2OS and its *POLQ* mutant (provided by Dr. J. Stark) [57] were grown and were irradiated as reported earlier [15,30]. *POLQ^−/−^* A549 cells were generated using CRISPR/Cas9^Nickase^. Two guide RNAs (gRNAs, Appendix A) targeting exon 2 were cloned into all-in-one plasmids (PX460) and delivered to the cells by nucleofection using the Nucleofector-2B device.

### 2.2. Cell Survival Determination

Cell survival was measured using the colony formation assay, as described previously [30].

### 2.3. RNA Interference

Knockdown experiments were carried out using specific siRNAs against the proteins of interest. The siRNAs (Appendix A) were delivered to the cells by nucleofection, as described previously [15,30].

### 2.4. Treatment of Cells with Inhibitors

All inhibitors (Appendix A) were added 1 h before irradiation unless indicated otherwise and were kept until the time of analysis.

### 2.5. Flow Cytometry (FC) Analysis of Resection

Resection analysis by FC through RPA70 detection was measured as described earlier [15,30]. A similar protocol was adopted for the analysis of resection through BrdU detection. In this case, BrdU at a concentration of 2 µM was added 24 h before irradiation in exponentially growing (Expo) cells or before serum deprivation (SD) in G_0_-phase cells. Antibodies used for this purpose are listed in Appendix A. Experiments were replicated three times and representative histograms from one experiment are shown.

### 2.6. FC Analysis of γH2AX Intensity

γH2AX intensity is measured by FC, as described earlier [58]. Experiments were replicated three times and representative histograms from one experiment are shown. Antibodies used for this purpose are listed in Appendix A.

### 2.7. Polyacrylamide Gel Electrophoresis (SDS-PAGE) and Western Blotting

SDS-PAGE and immunoblotting were employed, as previously described [15,30]. Antibodies used are listed in Appendix A.

### 2.8. Real-Time Polymerase Chain Reaction (PCR)

mRNA levels of CtIP were determined by real-time PCR. We used commercially available kits to extract total RNAs (Roche, 11828665001) and synthesize the first cDNA strands (Thermo fisher scientific, K1631). The PCR primers and conditions used are listed in Appendix A, respectively. Data were analyzed using Light-Cycler software Version 4.1 (Roche).

### 2.9. Pulsed-Field Gel Electrophoresis (PFGE)

To analyze repair of DSBs, in general, and specifically alt-EJ, PFGE was used, as described earlier [30]. The equivalent dose (Deq), rather than the fraction of DNA released (FDR), calculated using the corresponding dose–response curve, was adopted here as a parameter because it corrects the repair kinetics analysis for fluctuations in the dose–response curves. Six determinations from two independent experiments were used to calculate means and standard errors (SEs). Repair kinetics were fitted assuming two exponential components (fast and slow) of rejoining according to the equation, Deq = Ae^−bt^ + Ce^−dt^ [59], using a nonlinear regression analysis tool (Sigma-plot 14; Systat Software GmbH). The parameters A and C provide information on the fraction of DSBs processed by fast versus slow kinetics. The parameters b and d allow the calculation of the repair half times for the fast and the slow components of repair.

### 2.10. G_1_-Phase Cell Synchronization

A thymidine block combined with a nocodazole block was used to synchronize Expo cells in G_1_-phase. For details, see the legend of Appendix A.

### 2.11. Statistical Analyses

Results were expressed as mean ± standard error (SE) calculated from three or more repeats. Statistical significance between experimental groups was determined by using *t*-test. The significance of differences between individual measurements was indicated by using symbols: * *p* < 0.05, ** *p* < 0.01.

## 3. Results

### 3.1. Alt-EJ Is Suppressed as G_1_-Phase Cells Enter G_0_-Phase

We previously demonstrated using PFGE that alt-EJ is suppressed in mouse and hamster c-NHEJ mutants when they grow into a plateau- and enter G_0_-phase [31,60,61]. These results were later also confirmed using appropriately designed reporter assays [62]. Here, we extend these observations to human cells and begin with the elucidation of the underpinning mechanisms. Figure 1a shows that, under the conditions employed, 82-6 hTert cells grow logarithmically for 3 days and enter later, if not refed, a plateau phase. In the plateau phase, the majority of cells accumulate with a G_1_-phase-equivalent DNA content (Figure 1b, left panels). Figure 1b (right panel) shows that levels of Ki67, a widely used marker for discrimination between cells of G_1_- and G_0_ phases, decrease pronouncedly when cells enter plateau phase. These results are confirmed using Pyronin Y, an alternative marker of G_0_ cells (Appendix A). This shows that plateau-phase cultures mainly comprise G_0_-phase cells. 

To reduce irreproducibility associated with unfed plateau-phase cultures [15], we generated G_0_-phase 82-6 hTert cell populations using serum deprivation (SD) starting at day 2 of growth (Figure 1a). Figure 1a,b show good stability and enrichment in G_0_-phase in SD cultures, similar to that of plateau-phase cultures, at day 3. SD-cultures are, therefore, exclusively used in the following experiments. To compare the responses of G_0_- to those of G_1_-phase cells, we also employed the protocol outlined in Appendix A to obtain enriched populations of G_1_-phase cells. Appendix A shows that this protocol generates highly enriched G_1_-phase cells, 4 h after release from nocodazole block.

Analysis of DSB repair using PFGE after exposure to 20 Gy (Figure 1c, broken lines) shows indistinguishable repair potential in Expo, as well as in enriched G_0_- and G_1_-phase 82-6 hTert cells. Figure 1d summarizes the cell cycle distribution and growth state through PI and Ki67 staining. This repair activity mainly reflects the function of c-NHEJ [5,63], which removes over 90% of DSBs within 2 h, independently of the cell growth stage and cell cycle phase. The treatment of cells with the DNA-PKcs inhibitor NU7441 (2.5 µM, DNA-PKcsi) inhibits c-NHEJ and allows alt-EJ to come to the fore, since as mentioned above, HR (and SSA) is largely inactive in G_0_ and G_1_ cells [6]. This is also confirmed by the absence of RAD51 foci in these cells, despite efficient formation in G_2_-phase cells (Appendix A). In contrast, the formation of γH2AX foci increases linearly as a function of IR dose and shows the expected increase in G_2_-phase cells owing to their higher DNA content.

Exponentially growing cells treated with DNA-PKcsi show a profound reduction in DSB repair activity, but alt-EJ and, possibly, also HR and SSA repair over 50% of DSBs within 2 h; G_1_-phase cells show only slightly reduced DSB repair activity in the presence of DNA-PKcsi, which can mainly be attributed to alt-EJ, as HR and SSA are inactive. Additionally, the small difference in the kinetics between exponentially growing and enriched G_1_-phase cells that fails to reach statistical significance suggests that the contribution of HR and SSA in the former cells is small under the conditions employed here [5].

Strikingly, G_0_-phase cells treated with DNA-PKcsi are profoundly deficient in DSB repair, particularly at early stages after IR and repair less than 20% of induced DSBs in 4 h (Figure 1c). This is evidence for strong alt-EJ suppression, as cells transit from G_1_- to G_0_-phase of the cell cycle. Fitting of these curves to the sum of two exponentials, as described earlier [59], shows that in G_0_, the half time of repair (t50) of both the fast as well as the slow components of DSB rejoining increases (Appendix A) and that the contribution of the fast component is dramatically reduced. We conclude that the fast component of alt-EJ becomes strongly compromised as cells enter G_0_-phase.

Lung adenocarcinoma A549 cells can also be maintained under analogous growth conditions to generate highly enriched cultures of G_1_- and G_0_-phase cells (Appendix A). When DSB repair is analyzed in the presence of DNA-PKcsi, a similar, strong suppression of alt-EJ is observed in G_0_-phase cells (Appendix A). We conclude that suppression of alt-EJ in G_0_-phase cells is a general phenotype, detectable in several cell lines from different species [31,32,33], and focus below on the elucidation of the underpinning mechanisms.

To solidify our postulate that DSB rejoining in G_1_-phase cells after the inhibition of DNA-PKcs reflects mainly alt-EJ, we tested a mutant of A549 cells defective in the *POLQ* gene that encodes a key component of alt-EJ, Polθ [64]. The approach used to generate this mutant is described under “Materials and Methods”, and Figure 1e shows representative clones examined for a reduction in Polθ levels. We selected clone 8 for further experiments, and Figure 1f shows the DSB repair kinetics in *POLQ*-deficient and wild-type G_1_-A549 cells. In the absence of DNA-PKcsi, when c-NHEJ is fully active, deficiency in *POLQ* reduces repair only slightly (broken lines, less than a 5% reduction in any time point), with no statistical significance reached at any of the time points examined. Strikingly, in the presence of DNA-PKcsi, residual repair activity is significantly compromised in *POLQ^−/−^*A549 cells: 25% repair within 30 min and 50% repair within 2 h in wild-type A549 cells, but less than 20% repair in *POLQ^−/−^*A549 cells at 4 h after IR. Since Polθ is a component of alt-EJ, we surmise that residual repair activity after the inhibition of DNA-PKcs reflects the function of this pathway.

To confirm this postulate with a different cell line, we tested, in a similar experiment, a pair of U2OS cell lines proficient and deficient in Polθ [57]. The results obtained are summarized in Appendix A. Here, again, Polθ deficiency strongly compromises alt-EJ brought to the fore by the inhibition of DNA-PKcs. We also demonstrated earlier the dependence of this residual repair activity on LIG1/3 and PARP 1 [65,66], further confirming that, after c-NHEJ inhibition, repair in G_1_-phase mainly reflects the function of alt-EJ.

The results presented here and the majority of those reported earlier [31,60,61], analyze DSB repair at high IR doses. We recently reported profound adaptations in the DSB repair pathway choice with increasing IR doses up to 20 Gy [5,67,68,69]. We, therefore, examined the above-described suppression of alt-EJ in human G_0_-phase cells after exposure to 2 Gy, using analysis of γH2AX signal intensity by FC (Appendix A). We employed a modified method similar to that described previously [5,67,68,69] measuring the γH2AX signal in a cell-cycle-specific manner, as outlined in Appendix A.

A γH2AX signal reduction of nearly 50% within 6 h in untreated cells was observed (Appendix A), showing, as expected, that DSB repair can also be followed with this assay but that the kinetics measured are markedly slower than in the PFGE experiments discussed above. However, even with this assay, the repair kinetics of G_1_-phase and G_0_-phase cells are similar, with only a slight trend for slower repair kinetics in G_0_-phase cells (Appendix A). In the presence of DNA-PKcsi, the repair is not detectable in G_0_-phase cells using γH2AX analysis and becomes detectable in G_1_-phase cells only at 6 h post-IR (Appendix A).

We are currently investigating whether the slower repair kinetics and the reduced growth-state effect on alt-EJ at low IR doses reflects the earlier discussed [70] divergence between DSB repair analysis with assays measuring the physical DNA integrity versus the decay of the DDR protein γH2AX. We also explore whether larger differences become evident after longer periods of follow-up post-IR and whether they reflect the reported dose dependency of alt-EJ [71,72]. Therefore, in the experiments presented below, we focus on high-IR-dose effects.

### 3.2. DNA End-Resection Is Undetectable in G_0_-Phase Cells

As resection is believed to facilitate alt-EJ, we investigated whether reduced resection underpins its suppression in G_0_- versus G_1_-phase cells. Using methods similar to γH2AX detection, we measured RPA signals in cells in G_0_- and G_1_-phase.

Figure 2a,b show, as already reported [16], low-level resection in G_1_-phase cells of exponentially growing cultures after exposure to 10 Gy. Figure 2c confirms this low-level resection in G_1_ using BrdU signal analysis as an alternative to RPA for ssDNA detection and, thus, resection analysis, in irradiated cells [68]. In accordance with recent results in G_2_-phase [68,69], DNA-PKcsi fails to modify resection in G_1_-phase (Appendix A). Therefore, subsequent resection experiments were carried out in the absence of DNA-PKcsi. Similar patterns of resection were also observed for A549 cells, with low but detectable resection in G_1_-phase cells of exponentially growing cultures after exposure to 10 Gy (Appendix A), which remain unaffected by DNA-PKcsi (Appendix A).

Notably, similar analysis in G_0_-phase in both cell lines failed to detect resection (Figure 2a–c and Appendix A). These observations point to a direct link between reduced alt-EJ activity in G_0_- versus G_1_-phase cells and the associated reduction in resection activity. Because resection in G_1_-phase cells was relatively small (Figure 2a–c), we sought additional experimental validation of the putative connection between resection and alt-EJ activity by examining, under similar conditions, the changes in the enzymatic machinery of resection.

### 3.3. CtIP Levels Are Undetectable in G_0_-Phase Cells

The undetectable resection in G_0_-phase cells may derive either from a reduction in the abundance of components of the resection machinery or from a down-regulation of their activities. We, therefore, measured the levels of key proteins of the resection apparatus in exponentially growing, as well as in G_0_- and G_1_-phase 82-6 hTert cells, before and 1 or 3 h after exposure to 10 Gy. Figure 2d shows that while MRE11, RAD50, NBS1 and DNA2 show only minor fluctuations, CtIP and EXO1 are markedly reduced in G_0_-phase cells, as compared to G_1_-phase or exponentially growing cells. Irradiation profoundly increases the levels of CtIP in exponentially growing and G_1_-phase cells, but CtIP, as well as its radiation-induced increase, are undetectable in G_0_-phase cells. Because CtIP is a central regulatory component of the resection apparatus, its depletion in G_0_-phase cells explains the absence of resection.

The reduction in CtIP-protein levels in G_0_-phase cells may reflect the normally occurring reduction in gene expression during quiescence. We, therefore, measured the mRNA levels of CtIP in exponentially growing as well as in G_1_- and G_0_-phase cells. Figure 2e shows similar mRNA levels in the different conditions analyzed, thus excluding transcriptional regulation as a candidate mechanism.

We, therefore, turned our attention to CtIP stability. Figure 2f shows the successful depletion of CtIP, EXO1 and DNA2 in exponentially growing 82-6 hTert cells, and Figure 2g shows that CtIP and DNA2 depletion suppresses resection in G_1_-phase, whereas EXO1 depletion is ineffective. Additionally, the inhibition of MRE11 endonuclease activity with PFM01 inhibits resection, whereas Mirin, an inhibitor of MRE11 exonuclease activity [12], is ineffective (Figure 2g). Appendix A summarizes the results from several similar experiments, confirms the conclusions and shows their statistical power.

To directly connect resection with the function of alt-EJ in DSB repair, we analyzed G_1_-phase cells after depletion of CtIP, EXO1 or DNA2, or after the inhibition of MRE11 endonuclease or exonuclease activity. The results in Figure 3a show that CtIP knockdown not only suppresses resection profoundly, but also inhibits alt-EJ, whereas EXO1 knockdown, which has no effect on resection, leaves alt-EJ unaffected. In this experiment, DNA2 knockdown has no detectable effect on alt-EJ, which we interpret as evidence that short-range resection in G_1_-phase cells is sufficient for alt-EJ. Indeed, PFM01, which inhibits the endonuclease activity of MRE11 and is, thus, expected to specifically inhibit short-range resection, strongly suppresses alt-EJ, whereas Mirin fails to inhibit resection or alt-EJ.

We conclude that, in G_1_-phase cells, alt-EJ benefits from resection initiated by the MRN/CtIP and that the depletion of CtIP as cells enter G_0_-phase is causative to the suppression of alt-EJ observed. Combined with the results in Figure 1c, our observations further suggest that resection is required for fast DSB rejoining by alt-EJ. These observations, of course, immediately raise the question as to the mechanisms underpinning CtIP depletion in G_0_-phase cells.

### 3.4. APC^CDH1^ Depletes CtIP in G_0_-Phase Cells

APC^CDH1^ ubiquitinates CtIP and mediates its degradation by the proteasome in G_2_-phase [39,51,73]. We, therefore, examined the effect of the proteasome inhibitor bortezomib on CtIP levels in G_0_ cells. The treatment of G_0_-phase 82-6 hTert cells with 2 µM bortezomib for 2 h markedly increases CtIP levels (Figure 3b). Moreover, CDH1 knockdown similarly upregulates the expression of CtIP (Figure 3b). Finally, in A549 *CDH1^−/−^* cells [30], CtIP is clearly detectable in G_0_-phase and increases further somewhat in G_1_-phase cells. Interestingly, EXO1 levels remain unaffected by *CDH1* status and show the above-documented decrease in G_0_-phase (Figure 3c). These results convincingly show that increased APC/C^CDH1^ activity in G_0_-phase cells causes the degradation of CtIP.

We next investigated whether suppression of CtIP degradation using the above treatments restores alt-EJ and resection in G_0_-phase cells. Figure 3d,e show that CDH1 depletion, or treatment with bortezomib, markedly restores alt-EJ in G_0_-phase cells (Figure 3e), without affecting their cell cycle distribution (Appendix A). However, both *CDH1* depletion and Bortezomib failed to clearly elevate resection in G_0_-phase cells (Appendix A). We think this might be due to the relatively low sensitivity of the resection analysis method and the ability of alt-EJ to also function after only short-range resection of the DNA ends, as discussed above (see below for further discussion on the topic). Moreover, alt-EJ remains at G_1_-phase-levels when A549 *CDH1^−/−^* cells enter G_0_-phase (Figure 3f), as expected from the associated absence of CtIP degradation (Figure 3c). Collectively, these results identify APC/C as a regulator of alt-EJ activity operating by adjusting CtIP levels.

### 3.5. Low CDK Activity in G_0_-Phase Cells Keeps CtIP Inactive and alt-EJ Suppressed

CtIP activity is also regulated by CDK-dependent phosphorylations [35,36,37,38,39]. High CDK activity is a feature of proliferating cells and is low in G_0_-phase cells [74,75]. When mitotic cells divide and enter G_1_-phase, or when G_0_-phase cells are stimulated to proliferate by growth factors, CDK4/6 is activated by D-type cyclins, the expression of which is growth-factor-inducible [76]. In late G_1_-phase, Cyclin D-CDK4/6 complexes phosphorylate pocket proteins (RB1, p107 and p130) to release E2F transcription factors to induce the transcription of G_1_/S target genes, including the one encoding for cyclin E, thus activating CDK2/Cyclin E [77].

We considered the possibility that active CDK4/6 in G_1_-phase cells phosphorylates CtIP, and that this phosphorylation is important for CtIP activation and, thus, for resection and alt-EJ. We tested the effects of CDK inhibition on resection and alt-EJ in enriched G_1_-phase 82-6 hTert cells. Figure 4a shows that CDK4/6 inhibition, confirmed by the suppression of retinoblastoma protein (RB1) phosphorylation in serine 807 and 811 (Figure 4b), suppresses to G_0_ levels alt-EJ in G_1_-phase without affecting cell cycle distribution in a detectable manner (Figure 4c). On the other hand, the inhibition of CDK2 or CDK1 has no effect on DSB repair activity under these conditions. This can be explained by CDK4/6 being the sole CDK activity at this stage of G_1_-phase in non-transformed 82-6 hTert cells. Figure 4d shows that CDK4/6 inhibition also suppresses resection in G_1_-phase cells, whereas the effect of CDK2 or CDK1 inhibition fails to reach statistical significance. Figure 4e,f show that the elevation of CtIP levels by IR is suppressed by CDK4/6 inhibition, which suggests that the upregulation of CtIP in G_1_-phase cells induced by IR is CDK4/6-dependent. In addition, CDK4/6 inhibition also suppresses IR-induced CtIP phosphorylation at threonine 847 (CtIP-T847). These results suggest that CDK4/6-mediated stabilization, as well as CDK-mediated phosphorylation, contribute to the regulation of CtIP and resection in G_1_-phase cells.

It has been reported, when using confluent (practically G_0_-phase) 82-6 hTert cultures, that PLK3 phosphorylates and activates CtIP for resection to support a special form of resection-dependent c-NHEJ [78]. We, therefore, investigated the effects of PLK3 inhibition on the regulation of G_1_-phase alt-EJ and resection. PLK3 is known to be required for cell entry into S-phase by promoting the expression of cyclin E1 [79]. We confirmed PLK3 inhibition by analyzing cyclin E1 levels after the release of cells from G_0_ via transfer to a fresh growth medium in the presence or absence of the PLK1/3 inhibitor, GW843682X (2 µM). Appendix A shows that the levels of cyclin E1 strongly increase after 12 h in these cells and that GW843682X abrogates this increase. Notably, PLK3 inhibition fails to suppress alt-EJ (Appendix A) or resection (Appendix A) in G_1_-phase 82-6 hTert cells. Thus, PLK3 may be specifically utilized in G_0_-phase cells to regulate resection-dependent c-NHEJ, when CDK4/6 activity is low and alt-EJ is suppressed.

We investigated how the inhibition of CDKs affects alt-EJ in WT and *CDH1^−/−^* A549 cells, tested in G_1_- and G_0_-phase (Figure 5a). Notably, similar to 82-6 hTert cells, alt-EJ only depends on CDK4/6 in WT-enriched G_1_-phase A549 cells. In contrast, in *CDH1^−/−^* A549 cells, the single inhibition of CDK4/6, CDK1 or CDK2 exerts only small inhibitory effects on alt-EJ. However, a cocktail including all inhibitors causes a strong suppression of alt-EJ, both in enriched G_1_- as well as G_0_-phase cells (Figure 5a). As mentioned above, DSB induction upregulates CtIP levels in a CDK4/6-dependent manner in G_1_-phase-synchronized 82-6 hTert cells. Appendix A shows the same CtIP regulatory pattern in G_1_-phase A549 cells, which suggests that the upregulation of CtIP by IR in G_1_-phase cells is not specific for 82-6 hTert cells. However, the IR-dependent upregulation of CtIP is absent in *CDH1^−/−^* A549 G_1_ or G_0_-phase cells, and CDKsi also fails to affect CtIP levels detectably (Appendix A). Taken together, these results suggest that IR suppresses the activity of APC/C^CDH1^ in wild-type G_1_-phase cells in a CDK-dependent manner and, thus, upregulates CtIP.

The shift in the absolute dependence of alt-EJ on a specific set of CDKs likely reflects the general stabilization of cyclins in G_1_-phase following APC/C^CDH1^ inactivation in the above experiments, which leaves several CDKs active to interchangeably activate CtIP. Indeed, Figure 5b shows that in *CDH1^−/−^* A549 cells, cyclins A2 and B1 are stabilized, while the levels of cyclin D1 and cyclin E1 remain unaffected (Appendix A). The same trends are also observed in 82-6 hTert cells after CDH1 knockdown (Figure 5c and Appendix A). Appendix A should be changed.

Thus, *CDH1* depletion shifts CDK activity in early G_1_-phase from specific CDK4/6 activation to general CDK4/6, CDK1 and CDK2 activation, which explains the results obtained. Specifically, the degradation characteristics by APC/C of cyclins and CtIP in G_1_-phase allow CDK-dependent CtIP activation in the G_1_-phase of the cell cycle and, thus, the resection and alt-EJ activities observed. Notably, the results obtained with *CDH1^−/−^* cells also show that when CtIP is present in cells, its activity strictly requires CDK-dependent phosphorylation.

### 3.6. Suppression of Alt-EJ in G_0_-Phase Cells Enhances Radioresistance to Killing

The above results uncover the well-designed, programmed suppression of alt-EJ in G_0_-phase cells, which is part of the more general cell cycle regulation of CtIP levels and activity, which, in turn, regulate resection and alt-EJ. However, are there benefits for irradiated cells from such regulatory adaptations of the activities of specific DSB repair pathways?

It is relevant that alt-EJ is error-prone and, therefore, a putative major source of genomic instability. To address this question, we exposed cells to IR, either in G_0_- or G_1_-phase, and plated them at a lower density to allow for colony formation 6 h later. Figure 5d shows that, in both cell lines, G_0_-phase cells are significantly more radioresistant than G_1_-phase cells. In addition, in *CDH1^−/−^* A549 cells, G_0_-phase cells that retain resection and, therefore, alt-EJ are equally radiosensitive to G_1_-phase cells (Figure 5e). Notably, the suppression of alt-EJ by the inactivation of *POLQ* also compromises the differences in radiosensitivity between cells in G_1_- and G_0_-phase (Figure 5f). In addition, CDK4/6 inhibition renders A549 cells irradiated in G_1_-phase as radioresistant (Appendix A), providing further support to the concept and to the uncovered regulatory circuitry. We surmise that in G_1_-phase, cell survival benefits from the suppression of alt-EJ.

## 4. Discussion

### 4.1. CtIP Levels Regulate Resection and Alt-EJ throughout the Cell Cycle

We show above that CtIP cooperates with MRE11 to regulate resection in G_1_-phase cells. Indeed, CtIP is involved in the regulation of alt-EJ throughout the cell cycle in two distinct ways: First, CtIP levels fluctuate throughout the cell cycle and are, as we show here, undetectable in G_0_-phase phase owing to the activity of APC^CDH1^, which mediates nearly complete CtIP degradation. The inactivation of APC/C^CDH1^, as cells progress from G_0_- to G_1_- and then to S-phase [51,80], allows CtIP levels to increase, reaching a maximum at the end of G_2_-phase. Notably, our past and present work show that alt-EJ is almost fully suppressed in G_0_-phase and maximally active in G_2_-phase [31,32,33]. It remains to be investigated whether residual alt-EJ activity in G_0_-phase cells reflects residual CtIP activity or resection-independent alt-EJ proceeding with slow kinetics.

Second, CtIP activity is regulated by CDKs [36,37,38,39]. It is relevant, in this regard, that CDK activity is at its lowest level in quiescent cells, increases in G_1_-phase cells, and is again at maximum levels in G_2_-phase cells [81]. APC/C^CDH1^ activity marks cyclins for degradation, and, thus, suppresses CDK activity, which is required for CtIP activation [82]. This parallel evolution of ubiquitination and phosphorylation activities ensures that when CtIP is present, there will be CDK activity to prime it. It also explains why CDH1 deficiency or treatment with bortezomib has effects on both of these endpoints.

### 4.2. CtIP-Stability Regulation during the Cell Cycle and in Response to DNA Damage

HR is conceptually designed to faithfully restore the genome and to, thus, maintain genomic stability. CtIP-mediated resection exposes a 3′ single-stranded DNA that invades the sister chromatid during HR. In G2-phase cells, APC/C^CDH1^ mediates the clearance of CtIP from DSBs through ubiquitin-mediated degradation, thus limiting resection and adjusting it to the HR requirements [51]. We show here that CtIP enhances alt-EJ in G_1_-phase cells, thus promoting genomic instability (Figure 5d). CtIP is stabilized after IR in G_1_-phase cells (Figure 2d), which is opposite to the destabilization induced in G_2_-phase cells. This suggests benefits from CtIP stabilization, possibly at the expense of fidelity, at least when c-NHEJ is inactive. Indeed, *CtIP*-mutated chicken B cells *(*DT40) are sensitive to IR in G_1_-phase [83]. On the other hand, CtIP depletion rescues the DSB repair defect of *Artemis* mutants in G_1_-phase human fibroblasts but exerts only small effects on wild-type cells [16]. This observation suggests that CtIP depletion causes a switch from resection-dependent to resection-independent DSB processing in G_1_-phase cells [16]. Taken together, these data suggest that CtIP is involved in the repair of DSBs throughout the cell cycle, and that these specific functions and their regulation have a strong cell cycle component.

### 4.3. Relevance of Alt-EJ Suppression to Radiotherapy and Carcinogenesis

Processing of DSBs by alt-EJ generates chromosomal translocations that cause genomic instability and cancer [9]. DSBs naturally occur in a cell in the order of 10 to 50 DSBs per cell per day, depending on the cell cycle phase and tissue studied [84]. Most recent estimates put the number of cells in an average human body at around 37 trillion, most of which are non-cycling. These numbers indicate that a very high number of DSBs are induced every day in the human body. Therefore, protection from carcinogenesis may be achieved by suppressing the mutagenetic repair of spontaneous DSBs in the majority of cells.

On the other hand, tumor recurrence after radiation treatment is thought to be caused by radioresistant cells, which may derive from the pool of G_0_-phase cancer cells, which, as we show, are radioresistant. Here, the suppression of alt-EJ we show in quiescent cells might contribute to tumor recurrence by increasing the survival of tumor cells. Collectively, we, therefore, conclude that the default regulation of alt-EJ by CtIP suppresses carcinogenesis but may contribute to tumor recurrence. It will be important to develop means to harness this regulation in an effort to suppress carcinogenesis in normal cells or to decrease radioresistance in tumor cells.

### 4.4. Alt-EJ in G_1_-Phase Benefits from Resection When c-NHEJ Is Suppressed

The initiation of 5’-3’ resection at DNA ends is a critical determinant of the repair pathway choice, as it commits DSBs to HR while suppressing c-NHEJ [54,56,85]. Until recently, it was thought that resection only occurs in S- and G_2_-phase cells, and that it was inhibited in G_1_-phase by the Ku70/80 heterodimer and other proteins [54]. However, RPA loading at DSBs has, indeed, been observed in irradiated G_1_-phase cells, suggesting limited resection [56,86,87]. It has been suggested that in G_1_-phase, resection is dependent on the DSB number, with three or fewer endonuclease-induced DSBs per cell being insufficient and four or more being sufficient to activate resection [88]. Notably, resection has been reported in G_1_-phase human cells, particularly for complex DSBs induced by high LET particles or at high doses of X-rays [78,89]. Interestingly, 2 Gy of α-particles induced G_1_-resection-dependent DSB repair, which is independent of PARP1 and LIG1/3 but dependent on Artemis and DNA-PKcs. Because DNA-PKcs inhibition leaves the repair kinetics of Artemis deficient cells unchanged, it was concluded that this form of resection-dependent DSB repair reflects special forms of c-NHEJ rather than alt-EJ [16]. More work is needed to generate a mechanistic framework accommodating both sets of observations.

We previously demonstrated the suppression of alt-EJ when cells enter G_0_-phase [31,60]. Here, we extend these studies to human cells, uncover the role of resection in the regulation of alt-EJ and correlate the suppression of alt-EJ with compromised resection in G_0_-cells. Together, the above-discussed reports and our present results show that, on the one hand, resection can be induced in G_1_-phase, even in G_0_-phase cells by complex DSBs, to facilitate special forms of c-NHEJ [16]. On the other hand, simple DSBs of a c-NHEJ-deficient background activate resection specifically in G_1_-phase, which supports alt-EJ.

### 4.5. Collaboration between APC^CDH1^ and CDKs in the Regulation of DSB Repair in G_1_-Phase

CDK1 and CDK2 phosphorylate CtIP, NBS1, EXO1 and DNA2 to regulate resection [90,91,92] in G2-phase cells. Moreover, the necessity of CDK2-dependent CtIP phosphorylation on Thr847 is known for resection in G_1_-phase cells. However, here, using appropriately synchronized cells, we provide evidence for the contribution of CDK4/6 to CtIP activation and resection in G_1_-phase. Thus, it seems plausible that all cell cycle CDKs activate CtIP.

In addition to the phosphorylation-dependent regulation of CtIP activity, protein levels are also directly regulated and modulate resection in DSBs. APC/C^CDH1^ targets CtIP for proteasomal degradation after mitotic exit to generate the low resection environment that persists during G_1_-phase [51]. Moreover, it is well known that the oscillating activity of CDKs throughout the cell cycle is regulated by the periodic degradation of cyclins and CDK inhibitors (CKIs) by the ubiquitin–proteasome system [48], in which the APC/C^CDH1^ complex plays a central role, as it targets cyclin B1 and cyclin A2 for degradation. On the other hand, the affinity of CDH1 toward APC/C is suppressed by CDK phosphorylation [49,50]. These feedback mechanisms define the crosstalk between APC^CDH1^ and CDKs in the regulation of the cell cycle with extensions to DSB repair. These findings place the ubiquitination pathway and, particularly APC^CDH1^, in an intermediate node in the DSB repair pathway choice network.

## Figures and Tables

**Figure 1 cells-12-01530-f001:**
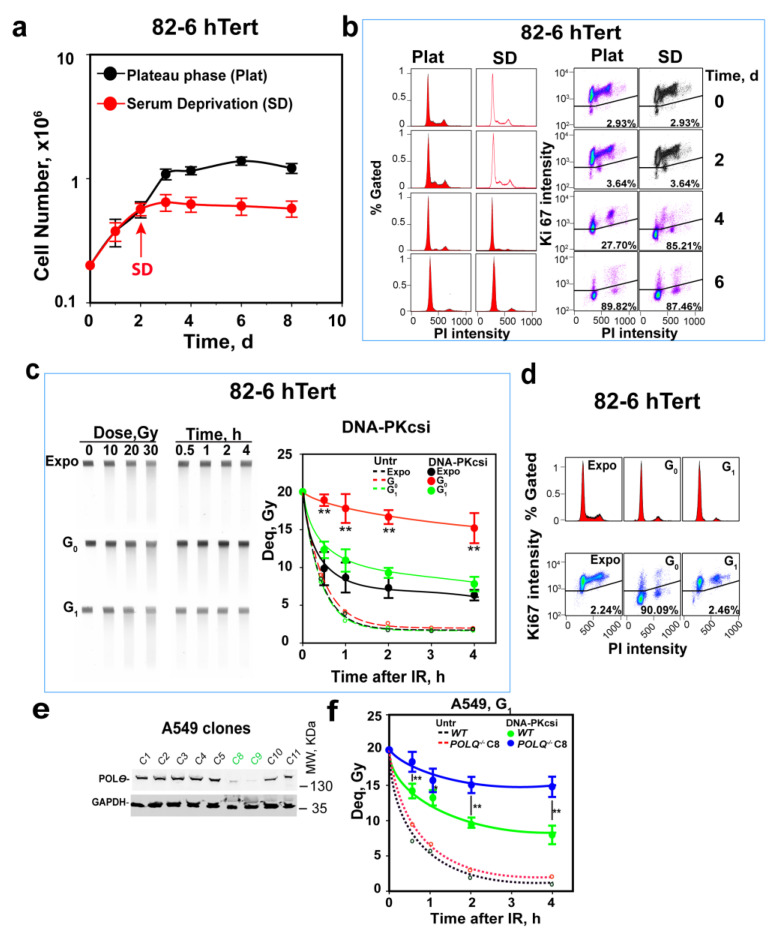
Suppression of alt-EJ in G_0_-phase cells. (**a**) Proliferation of 82-6 hTert cells under normal growth conditions, as well as after transfer to SD-medium two days later. Plot shows mean ± SE from 3 independent experiments. (**b**) Cell cycle distribution and cell growth status determination by PI combined with Ki67 staining at the indicated times of growth. All experiments are repeated 3 times, and a representative one is shown. (**c**) Kinetics of DSB repair measured by PFGE after exposure of cells to 20 Gy in the presence or absence of DNA-PKcsi, NU7441, at a concentration of 2.5 µM. The image on the left shows a typical PFGE gel with the dose–response and the repair kinetics analysis in the presence of DNA-PKcsi for exponentially growing as well as G_0_ and G_1_-enriched cells. The plot on the right shows results obtained by densitometry analysis of six similar gels in two experiments. Broken lines show results obtained with untreated cells and are shown without error bars to avoid congestion in the figure. The solid lines reflect the repair kinetics measured in the presence of DNA-PKcsi and are generated by fitting to the sum of two exponentials, as described under Materials and Methods. (**d**) PI and Ki67 staining of cells used in (**c**) generated to confirm the growth status of the cells used. (**e**) Western blot validation of *Polθ* depletion in selected A549 clones. Clones C8 and C9 show clear depletion of the protein. (**f**) Kinetics of DSB repair after exposure to 20 Gy in the presence or absence of DNA-PKcsi in *POLQ* wild-type and mutant (clone 8) G_1_-phase A549 cells. Plot shows mean ± SE of 6 determinations from 2 independent experiments. Untr, Expo, DNA-Pkcsi, WT and Deq represent untreated, exponentially growing, DNA-PKcs-inhibitor-treated, wild-type and dose equivalent, respectively; Symbols ** in (**c**,**f**) indicate *p* < 0.01 in statistical significance calculation, symbols * indicate *p* < 0.05.

**Figure 2 cells-12-01530-f002:**
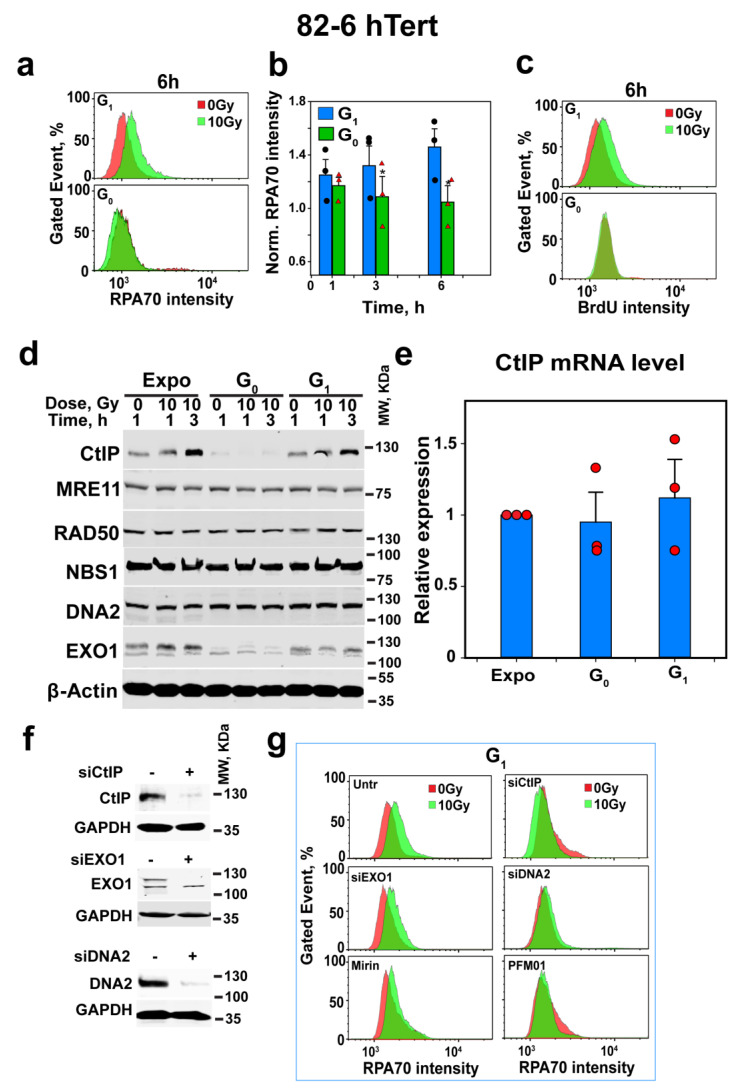
Depletion of CtIP as cells enter G_0_-phase suppresses resection. (**a**) Representative analysis of resection at DSBs, specifically in G_1_-phase cells discriminated by using PI- and EdU-combined staining after exposure to 10 Gy using RPA signal intensity as parameter and quantification by FC in 82-6 hTert cells; for details, see under “Material and Methods”. (**b**) Combined results from three independent experiments similar to those shown in (**a**) as a bar plot showing the normalized RPA70 signal intensity, as well as the associated SE and the significance of the observed differences. Black circles and red triangles show the results of individual experiments used in the analysis. Symbol * indicates *p* < 0.05 in statistical significance calculation. (**c**) As in (**a**), but for BrdU signal intensity to label single-stranded DNA and analyze resection using an alternative method to RPA staining. Experiments were repeated 3 times, and a representative one is shown. (**d**) WB analysis showing the levels of CtIP, MRE11, RAD50, DNA2 and EXO1 in 82-6 hTert cells in different stages of growth and phases of the cell cycle, measured at different times after exposure to 0 Gy or 10 Gy of IR. Experiments were repeated 3 times and a representative one is shown. (**e**) mRNA levels of CtIP in 82-6 hTert cells at different stages of growth and phases of the cell cycle. Red circles show the results of individual experiments used in the analysis Plot shows mean ± SE from 3 independent experiments. (**f**) Western blot analysis of CtIP, EXO1 and DNA2 after 48 h protein knockdown using specific siRNAs, which was conducted to confirm the efficient depletion of targeted protein for the experiment shown in (**g**). (**g**) Resection analysis as in (**a**) after knockdown or inhibition of CtIP, MRE11, EXO1 or DNA2 in G_1_-phase 82-6 hTert cells.

**Figure 3 cells-12-01530-f003:**
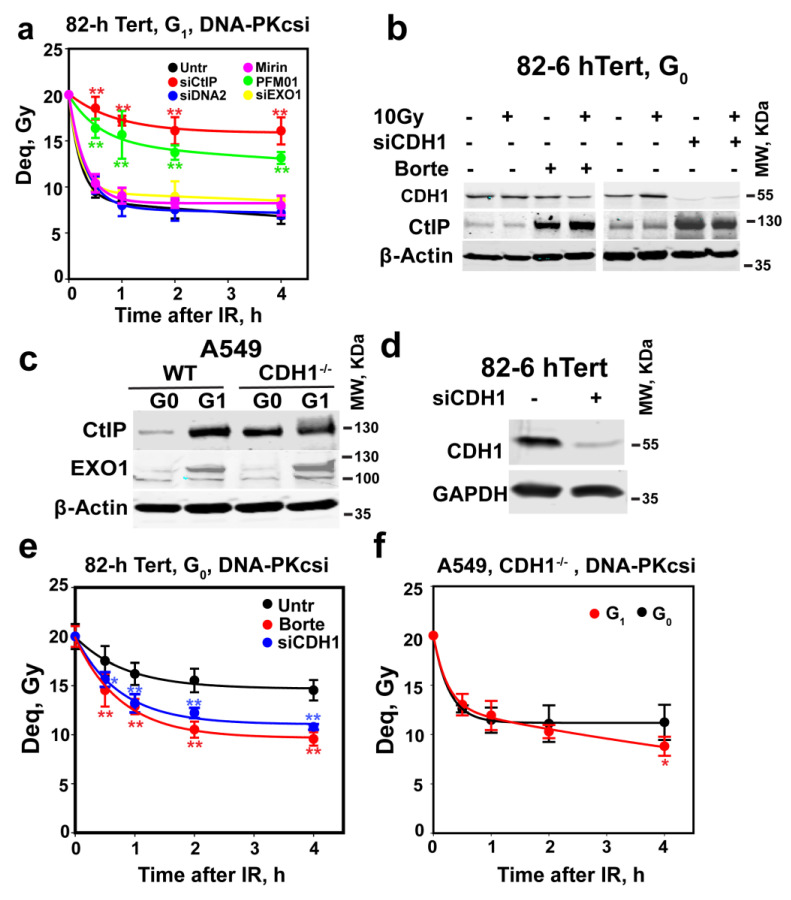
Depletion of CtIP suppresses alt-EJ in G_0_-phase cells. (**a**) As in Figure 1c for 82-6 hTert cells exposed to 20 Gy and maintained in the presence of DNA-PKcsi after knockdown or inhibition of CtIP, MRE11, EXO1 and DNA2. Plot shows mean ± SE from 6 determinations from 2 independent experiments. (**b**) WB analysis of CtIP levels in cells treated with 2 µM bortezomib, or depleted of CDH1 using specific siRNAs, in G_0_-phase 82-6 hTert cells. Bortezomib was added 2 h before IR and kept for 3 h after IR. (**c**) WB analysis of CtIP and EXO1 in *CDH1* wild-type and mutant G_0_- or G_1_-phase A549 cells; all experiments were repeated 3 times in (**b**,**c**), and a representative one is shown. (**d**) WB analysis of CDH1 knockdown in cells used in (**e**). (**e**) Rescue of alt-EJ in G_0_-phase 82-6 hTert cells after treatment with bortezomib for 2 h, or 48 h CDH1 knockdown, as shown in (**d**). (**f**) Alt-EJ in *CDH1^−/−^* A549 cells in G_0_- and G_1_-phase of the cell cycle. Borte in (**b**) represents bortezomib. Plots in (**e**,**f**) show mean ± SE of 6 determinations from 2 independent experiments. Symbol ** in (**a**,**e**) indicates *p* < 0.01 in statistical significance calculation, symbols * indicate *p* < 0.05.

**Figure 4 cells-12-01530-f004:**
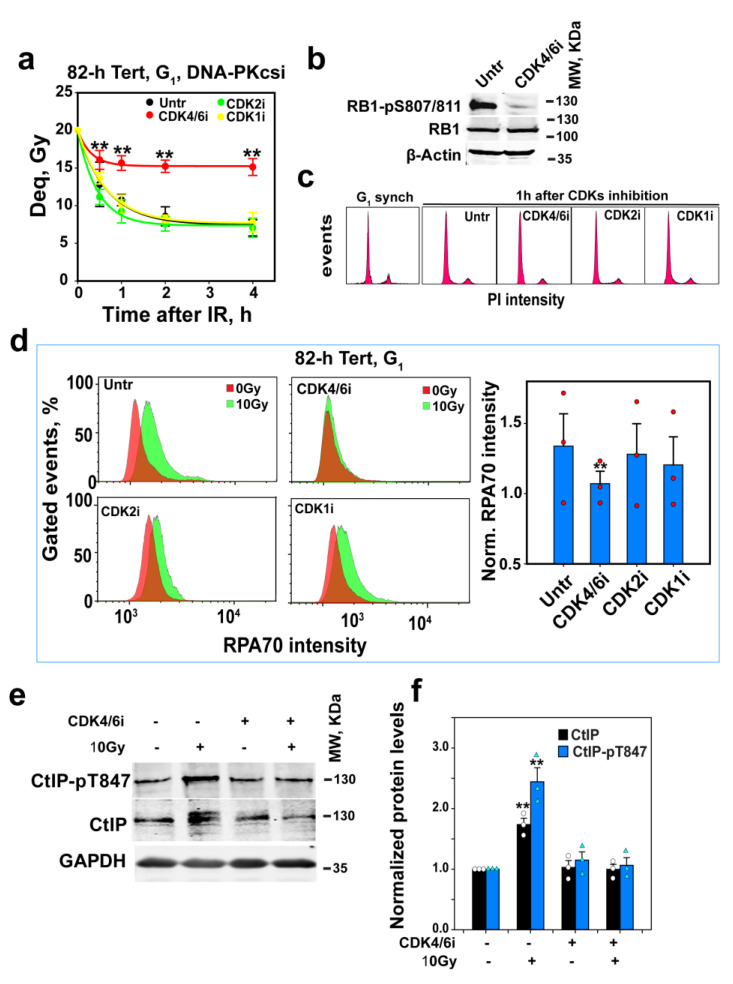
CDK4/6 inhibition suppresses alt-EJ and resection in G_1_-phase cells. (**a**) As in Figure 3a after inhibition of CDK4/6, CDK2 or CDK1 in G_1_-phase 82-6 hTert cells; CDK4/6i, CDK2i and CDK1i were added 1 h before IR at a concentration of 500 nM, 5 µM and 10 µM, respectively. Plot shows mean ± SE from 6 determinations in 2 independent experiments. (**b**) WB analysis of RB1-pS807/811 in G_1_-phase 82-6 hTert cells to confirm the efficacy of CDK4/6 inhibition by PD032991. Cells were treated with PD032991 1h before collection. Experiments were repeated 3 times, and a representative one is shown. (**c**) Cell cycle distribution of cells before and after CDK inhibition used in (**a**). (**d**) Resection analysis, as in Figure 2a, after CDK4/6, CDK2 or CDK1 inhibition in 82-6 hTert cells in G_1_-phase. Red circles show the results of individual experiments used in the analysis. Plot on the right shows mean ± SE from 3 independent experiments. (**e**) WB analysis of the effect of CDK4/6 inhibition on CtIP and CtIP-pT847 3h after exposure to 0 Gy or 10 Gy in G_1_-phase 82-6 hTert cells. Experiments were repeated 3 times, and a representative one is shown. (**f**) Measurements of band density and statistical analysis of results shown in (**e**). White circles and turquoise triangles show the results of individual experiments used in the analysis Plot shows mean ± SE from 3 independent experiments. Symbol ** in (**a**,**d**,**f**) indicates *p* < 0.01 in statistical significance calculation.

**Figure 5 cells-12-01530-f005:**
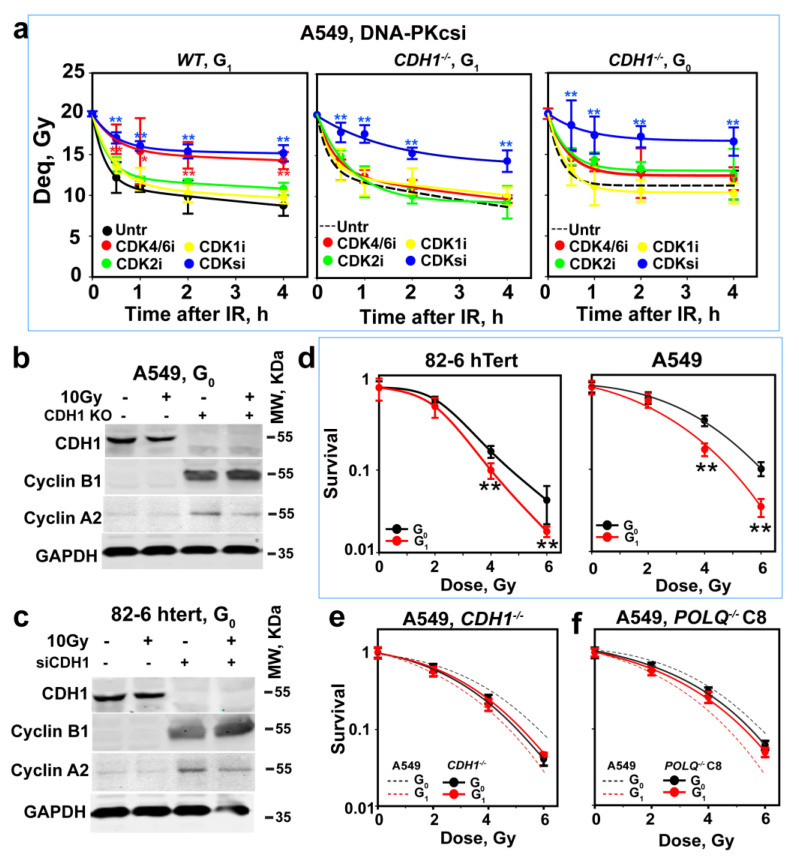
CDH1 depletion rescues alt-EJ after CtIP depletion but also alters the CDK dependence of CtIP activation. (**a**) As in Figure 3a after inhibition of CDK4/6, CDK2 or CDK1, alone or combined, in G_1_- or G_0_-phase wild-type or *CDH1^−/−^* A549 cells. Plot shows mean ± SE from 6 determinations in 2 independent experiments. (**b**) WB analysis showing the effects of *CDH1* depletion on cyclins A2 and B1 in G_0_-phase A549 cells. Experiments were repeated 3 times, and a representative one is shown. (**c**) As in (**b**) for 82-6 hTert cells. (**d**) Radiosensitivity of G_0_- and G_1_-phase 82-6 hTert and A549 cells determined using colony formation, as described in Material and Methods. (**e**) As in (**d**) for *CDH1^−/−^*A549 cells and their WT counterparts (broken lines). (**f**) As in (**e**) for *POLQ^−/−^* A549 cells, clone 8, and their WT counterparts (broken lines). Plots in (**d**–**f**) show mean ± SE from 3 independent experiments. Symbol * and ** in (**a**,**d**) indicate *p* < 0.05 and *p* < 0.01, respectively, in statistical significance calculation.

## Data Availability

As requested.

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
