# Peer review of "Low CDK Activity and Enhanced Degradation by APC/C^CDH1^ Abolishes CtIP Activity and Alt-EJ in Quiescent Cells"

_cells, 2023, doi:10.3390/cells12111530_

Round 1

Reviewer 1 Report

Manuscript entitled ‘Low CDK activity and enhanced degradation by APC/CCDH1 abolishes CtIP activity and alt-EJ in quiescent cells’ by Li et al. reported that CtIP is required for alt-EJ activity in G1-phase and that its degradation or reduced phosphorylation abrogates resection and alt-EJ, and proposed that the suppression of error-prone alt-EJ is a natural mechanism safeguarding genomic stability and protecting against tumorigenesis.

The focus of this manuscript is very interesting and novel which deserves further study. However, the project construction and manuscript description raise several major issues.

1. Authors intended to build two different models to obtain the cell populations at G0 and G1 phase, respectively. This is a key experiment for this project, and G0/G1 are hard to separate. However, a solid result to prove these two different populations are absent. 

In figure 1, authors showed data to prove cells are at G0 phase using 82-6hTERt cells, but data to show cells at G1 is not enough. How to tell the two different populations? Only by Ki67? Authors should also perform Hoechst 33342 and Pyronin Y Staining (Flow Cytometric Detection of G0 in Live Cells by Hoechst 33342 and Pyronin Y Staining), and data to prove cells at G0 or G1 phase should be compared side by side using two cell lines. 

Using ki67 as marker, except Flow cytometry, authors should use another method to prove these two populations, like PCR, WBs. 

2. For section of 3.5 Low CDK activity in G0-phase cells keeps CtIP inactive and alt-EJ suppressed.

Authors inhibit CDK4/6, however, no details to show the inhibit effects (WBs or PCR). High concentration of siRNA will as well inhibit cell growth, however, no data to show the growth rate of the cells depleted of CDK4/6. Authors intend to get the conclusion that CDK4/6 mediated phosphorylation of CtIP-T847 is part of the mechanism regulating resection in G1-phase cells by figure 4d. However, the ratio between phosphorylation of CtIP-T847/total CtIP seems not differ much. The conclusion is weak here. 

3. Most figures in this manuscript are not described clearly, such as i) no MW for each protein shown by WBs; ii) Figure 1c, ‘The image on the left shows a typical gel’, what does it mean? iii), Figure 1c, 1f, 5e and 5f, no data point for the dash line? iv), what is Expo, Borte, Deq, IR. Un.tr?

4. Description need to be precise.  In the absence of DNA-PKcsi when c-NHEJ is fully active, deficiency in POLQ only has a very small (what is the number?) reduction on repair (broken lines). Strikingly, in the presence of DNA-PKcsi, residual repair activity, is strongly compromised (how strong?) in POLQ-/-A549 cells. In addition, we have shown before the dependence of this residual repair on LIG1/3 and PARP 1[61,62], confirming that after c-NHEJ inhibition, repair in G1–phase reflects the function of alt-EJ.

Author Response

Reviewer 1

Manuscript entitled ‘Low CDK activity and enhanced degradation by APC/CCDH1 abolishes CtIP activity and alt-EJ in quiescent cells’ by Li et al. reported that CtIP is required for alt-EJ activity in G1-phase and that its degradation or reduced phosphorylation abrogates resection and alt-EJ, and proposed that the suppression of error-prone alt-EJ is a natural mechanism safeguarding genomic stability and protecting against tumorigenesis.

The focus of this manuscript is very interesting and novel which deserves further study. However, the project construction and manuscript description raise several major issues.

Response: We greatly appreciate the positive comments of the Reviewer on the novelty of the manuscript.

  1. Authors intended to build two different models to obtain the cell populations at G0 and G1 phase, respectively. This is a key experiment for this project, and G0/G1 are hard to separate. However, a solid result to prove these two different populations are absent. 

In figure 1, authors showed data to prove cells are at G0 phase using 82-6hTERt cells, but data to show cells at G1 is not enough. How to tell the two different populations? Only by Ki67? Authors should also perform Hoechst 33342 and Pyronin Y Staining (Flow Cytometric Detection of G0 in Live Cells by Hoechst 33342 and Pyronin Y Staining), and data to prove cells at G0 or G1 phase should be compared side by side using two cell lines. 

Using ki67 as marker, except Flow cytometry, authors should use another method to prove these two populations, like PCR, WBs. 

Response: We fully agree with the reviewer that separation between G1 and G0 cells is important for the content and the concepts advance in our paper. Indeed, we addressed this important point in a previous publication [1] and found that changes of Ki67 and Pyronin Y between G1 and G0 cells show very similar trends. We carried out these experiments with several cell lines and have numerus repeats. From these experiments we learned that both methods give similar results and assign very similar subpopulations to G1 versus the G0 phase. Importantly, these experiments also showed that Ki67 gives a better signal that ensures a superior dynamic range for analyzing differences between G0 and G1 cells and was therefore adopted exclusively in the present manuscript. There is no universally accepted single method to definitively discriminate G0 from G1 cells, but Ki67 is the most widely used. We discuss in more detail these experiments and the equivalence of outcome in the revised manuscript.

Furthermore, we would like to point out that the cell systems used in our studies (82-6 hTert etc.) are among the most widely used and reliable systems for generating G0 or G1 phase cells. Indeed, there are numerous publications on the topic [1-4].

We really appreciate the concrete suggestions of the Reviewer for additional methods to separate G0 and G1 phase cells and plan testing them along the established ones in future work. However, we feel that the above response and the clarifications given address the key concern of the Reviewer: We have two methods and use cell lines that are widely used to generate G0 cells.

The field is highly competitive and we are eager to publish the present report as soon as possible in order to be one of the first that advance these concepts. This will be helpful not only for our future funding efforts, but also for “Cells”. 

  1. For section of 3.5 Low CDK activity in G0-phase cells keeps CtIP inactive and alt-EJ suppressed. Authors inhibit CDK4/6, however, no details to show the inhibit effects (WBs or PCR). High concentration of siRNA will as well inhibit cell growth, however, no data to show the growth rate of the cells depleted of CDK4/6. Authors intend to get the conclusion that CDK4/6 mediated phosphorylation of CtIP-T847 is part of the mechanism regulating resection in G1-phase cells by figure 4d. However, the ratio between phosphorylation of CtIP-T847/total CtIP seems not differ much. The conclusion is weak here. 

Response: Thanks for pointing out these relevant issues. To the first point: A well-known target of CDK4/6 is the retinoblastoma protein RB1, which inhibits cell-cycle progression until its inactivation by phosphorylation [5]. We therefore carried out additional experiments to test the effects of CDK4/6 inhibitor on the phosphorylation of RB1 at Serin 807/811. We sumarize the results obtained in Figure 4b of the revised manuscript, which confirm the postulated inhibition.

To the second point that high siRNA concentration and long-time treatment with inhibitors will inhibit cell growth: In the present study we use 1 h preirradiation treatment with the inhibitor and observation times after IR that are unlikely to alter cell cycle distribution. Indeed, we show the cell cycle distribution before and 1h after specific inhibitors treatment in Figure 4c. The results indicate that under the conditions used in our study, effects on cell growth remain undetectable. 

It is true that CDK4/6 inhibition does not seem to affect the ratio between phosphorylated versus total CtIP. We appreciate reviewer’s observation on this important point. IR at 10 Gy increases not only the phosphorylated level of CtIP, but also its total level in the absence of CDK4/6 inhibitor. Moreover, CDK4/6 inhibition suppresses the increase of both forms of the protein. This result suggests two levels of CtIP regulation: protein levels and phosphorylation. In the revised manuscript we have adapted our interpretation to consider this fact.

  1. Most figures in this manuscript are not described clearly, such as i) no MW for each protein shown by WBs; ii) Figure 1c, ‘The image on the left shows a typical gel’, what does it mean? iii), Figure 1c, 1f, 5e and 5f, no data point for the dash line? iv), what is Expo, Borte, Deq, IR. Un.tr?

Response: We appreciate the Reviewer pointing out these deficiencies. In the revised version of the manuscript we have extended the presentation and discussion of the individual figures and add more information useful to the reader, including all aspects kindly pointed out by the Reviewer.

  1. Description need to be precise.  In the absence of DNA-PKcsi when c-NHEJ is fully active, deficiency in POLQ only has a very small (what is the number?) reduction on repair (broken lines). Strikingly, in the presence of DNA-PKcsi, residual repair activity, is strongly compromised(how strong?) in POLQ-/-A549 cells. In addition, we have shown before the dependence of this residual repair on LIG1/3 and PARP 1[61,62], confirming that after c-NHEJ inhibition, repair in G1–phase reflects the function of alt-EJ.

Response: Again, thanks to the Reviewer for pointing out this weakness. In the Revision we describe our results in a much more quantitative manner.

Literature cited in the Response to Reviewers

  1. Li, F.; Mladenov, E.; Dueva, R.; Stuschke, M.; Timmermann, B.; Iliakis, G. Shift in G1-Checkpoint from ATM-Alone to a Cooperative ATM Plus ATR Regulation with Increasing Dose of Radiation. Cells 2022, 11, 63.
  2. Biehs, R.; Steinlage, M.; Barton, O.; Juhasz, S.; Kunzel, J.; Spies, J.; Shibata, A.; Jeggo, P.A.; Lobrich, M. DNA Double-Strand Break Resection Occurs during Non-homologous End Joining in G1 but Is Distinct from Resection during Homologous Recombination. Mol Cell 2017, 65, 671-684 e675, doi:10.1016/j.molcel.2016.12.016.
  3. Barton, O.; Naumann, S.C.; Diemer-Biehs, R.; Künzel, J.; Steinlage, M.; Conrad, S.; Makharashvili, N.; Wang, J.; Feng, L.; Lopez, B.S.; et al. Polo-like kinase 3 regulates CtIP during DNA double-strand break repair in G1. Journal of Cell Biology 2014, 206, 877-894, doi:10.1083/jcb.201401146.
  4. Averbeck, N.B.; Ringel, O.; Herrlitz, M.; Jakob, B.; Durante, M.; Taucher-Scholz, G. DNA end resection is needed for the repair of complex lesions in G1-phase human cells. Cell Cycle 2014, 13, 2509-2516, doi:10.4161/15384101.2015.941743.
  5. Topacio, B.R.; Zatulovskiy, E.; Cristea, S.; Xie, S.; Tambo, C.S.; Rubin, S.M.; Sage, J.; Koivomagi, M.; Skotheim, J.M. Cyclin D-Cdk4,6 Drives Cell-Cycle Progression via the Retinoblastoma Protein's C-Terminal Helix. Mol Cell 2019, 74, 758-770 e754, doi:10.1016/j.molcel.2019.03.020.

Reviewer 2 Report

The authors performed experiments to test DSB processing and repair in cells of two different human cell lines accumulated in G1 or in G0 (quiescent) by depletion of serum. The obtained results match perfectly with current knowledge that DSB processing and accumulation of ssDNA covered by RPA are greatly reduced in G1 cells and nearly undetectable in G0 cells. It is also showed that POLQ-dependent Alt-EJ repair can be addressed by PFGE approach in G1 cells treated with an inhibitor of the DNA-PKcs to block NHEJ, which is the principal DSB repair pathway in the experimental condition tested. Further, the authors showed very low levels of CtIP in G0 cells due to CDH1-dependent protein degradation by the proteasome. Additional results by specific inhibitors also indicated that the CtIP activity involved in Alt-EJ in G1 cells is promoted by CDK4/CDK6-dependent phosphorylation. Finally, it is shown that the inhibition of Alt-EJ in G1 cells leads to increased IR resistance to the level observed in G0 cells.

Interestingly, the authors proposed that the suppression of mutagenic Alt-EJ events is critical to preserve genome stability in non-cycling cells of eukaryotes organisms.

General comment: The experiments are presented and discussed in a very clear way. The manuscript is well written, and presentation is fluent and easy to follow. In my opinion, there are results that do not represent real novelty in the field, but I appreciate the way and clarity of their presentation. A weakness of the work is the lack of a direct and strong conclusion on CtIP by the expression of specific phosphorylation variants.

Specific concerns:

1) In the materials and methods, the procedure to analyze DSB repair by PFGE is not described and is difficult to understand what exactly is quantified in the Deq/Time graphs. 

2) Fig 1e-f. It is recommended to test also the other POLQ -/- clone (C9). The same comment for the viability test in Figure 5f.

3) Fig S1c. It is important to show gammaH2AX as a marker of DSB formation. However, I think that to make stronger point on HR in G0 cells, RAD51 and gammaH2AX foci in G0 treated cells in respect with the G1 must be shown. 

4) Fig 2d. Comment on the protein levels: it is curious that only CtIP and EXO1 levels are controlled in G1/G0 transition, while all the other factors involved in resection are not. Do you think that the activities of MRE11 complex and DNA2 are restrained by different mechanisms in G0 or, rather, still active to do some works? 

Another curiosity: Is POLQ present in G0 cells treated by IR?

5) Fig.3a. It is not so clear to me why depletion of DNA2 does not affect repair by Alt-EJ if resection is limited (Fig 2g). Is it possible that marginal contribution of DNA2 can be evident only in special conditions, such as cells carrying mutation in other factors involved in resection? For instance, what about resection and repair in double depleted DNA2+EXO1? Or DNA2+PFM01 (MRE11-endo)? 

6) Fig3b-d. As informative control, can you show EXO1 levels in CDH1 depleted cells?

7) Fig3e-f. As additional control, to rule out EXO1 and/or DNA2 contributions it will be nice to show repair profile in siCDH1 cells carrying depletion of EXO1 or DNA2. 

8) Fig4d. I’m confused about CtIP stability in CDK inhibited cells. Can you show CtIP levels in G1/G0 cells treated with the different inhibitors for CDK1/2/4/6? Related to this and Fig5a, how is CtIP levels and resection profile in CDKsi treated cells? 

9) Fig.5f. I think that it will be very interesting to try the same viability assay in cells treated with the CDK inhibitors.

10) Fig S5. How can we be sure that PLK1/3 are inhibited in the experiments?

Typos:

-lines 255 and 256: gammaH2AX is written with a wrong symbol/letter. 

- line 442: POLQ, without the “minus” apex.

Author Response

Please see the authors' response in the attached file.

Round 2

Reviewer 1 Report

Manuscript entitled ‘Low CDK activity and enhanced degradation by APC/CCDH1 abolishes CtIP activity and alt-EJ in quiescent cells’ by authors Li et al.  aimed to establish cell models showing cells at either G0 or G1 phase, identify CtIP-dependent resection as the key regulator and report that CtIP activation in G0-phase cells requires CDK-dependent phosphorylation by any available CDK, but is restricted to CDK4/6 at the early stages of the normal cell cycle. 

Two major issues

1. The confirmation of G0 and G1 cells

Authors clearly and nicely described the way to establish cells at G0 phase. However, the evidence to differ G0 and G1 cells is weak. Authors compared the Ki67 density in these cells and claimed that ‘We have reported that changes of Ki67 and Pyronin Y in G1 and G0 cells show very similar trends. However, Ki67 gives a better signal that ensures a superior dynamic range for analyzing differences [15]’. The authors still need to do Pyronin Y(present the data in the same manner as in Figure 1d) and show that consistent results were noticed using two different methods. After this, the authors can select Ki67 for the rest of the experiments.

2. Low CDK activity in G0-phase cells keeps CtIP inactive and alt-EJ suppressed

The authors aimed to show that inhibit CDK4/6, CtIP inactive and cells are more radioresistant. Figure 4b did not mention how long the cells were treated with CDK inhibitors. It has been reported that the half-life of CDK4 is 5 h1. The authors need to demonstrate that this complete inhibition of CDK4/6 can be achieved after one hour. Figure 4E requires measurement of band density and statistical analysis, it is difficult to conclude that CtIP levels are elevated by IR based on the plot in 4E.

Minor comments: 

1. Figure 1d add X and Y-axis.

2. Figure 2b, 2e, 4d, using bar chart with dots.

1          Wang, H., Goode, T., Iakova, P., Albrecht, J. H. & Timchenko, N. A. C/EBPalpha triggers proteasome-dependent degradation of cdk4 during growth arrest. EMBO J 21, 930-941 (2002). https://doi.org:10.1093/emboj/21.5.930

Author Response

Reviewer 1

Manuscript entitled ‘Low CDK activity and enhanced degradation by APC/CCDH1 abolishes CtIP activity and alt-EJ in quiescent cells’ by authors Li et al.  aimed to establish cell models showing cells at either G0 or G1 phase, identify CtIP-dependent resection as the key regulator and report that CtIP activation in G0-phase cells requires CDK-dependent phosphorylation by any available CDK, but is restricted to CDK4/6 at the early stages of the normal cell cycle. 

Two major issues

  1. The confirmation of G0 and G1 cells

Authors clearly and nicely described the way to establish cells at G0 phase. However, the evidence to differ G0 and G1 cells is weak. Authors compared the Ki67 density in these cells and claimed that ‘We have reported that changes of Ki67 and Pyronin Y in G1 and G0 cells show very similar trends. However, Ki67 gives a better signal that ensures a superior dynamic range for analyzing differences [15]’. The authors still need to do Pyronin Y (present the data in the same manner as in Figure 1d) and show that consistent results were noticed using two different methods. After this, the authors can select Ki67 for the rest of the experiments.

Response: We show now the required data in Figure S1c. 

  1. Low CDK activity in G0-phase cells keeps CtIP inactive and alt-EJ suppressed

The authors aimed to show that inhibit CDK4/6, CtIP inactive and cells are more radioresistant. Figure 4b did not mention how long the cells were treated with CDK inhibitors. It has been reported that the half-life of CDK4 is 5 h1. The authors need to demonstrate that this complete inhibition of CDK4/6 can be achieved after one hour. Figure 4E requires measurement of band density and statistical analysis, it is difficult to conclude that CtIP levels are elevated by IR based on the plot in 4E.

Response:

  • Since cells are treated with sufficiently high concentrations of PD032991 all available molecules in the cell are expected to be inhibited. As a consequence, the half life of the relevant CDKs is not a contributing parameter here. We give the treatment time used in this experiment in the revised version of the paper, as requested by the Reviewer.

  • We added the required measurements and their statistical analysis in Figure 4f.

Minor comments: 

  1. Figure 1d add X and Y-axis.

Response: Done.

  1. Figure 2b, 2e, 4d, using bar chart with dots.

Response: Done.

Reviewer 2 Report

I have appreciated the efforts done by the authors to address my concerns. I think that the manuscript is improved and relevant experimental controls are now present. 

Author Response

No additional comments.